# An evidence based efficacy and safety assessment of the ethnobiologicals against poisonous and non-poisonous bites used by the tribals of three westernmost districts of West Bengal, India: Anti-phospholipase A$_2$ and genotoxic effects

Biplob Kumar Modak[1], Partha Gorai[1], Devendra Kumar Pandey[2], Abhijit Dey[3]*, Tabarak Malik[4]*

1 Department of Zoology, Sidho-Kanho-Birsha University, Lagda, West Bengal, India, 2 Department of Biotechnology, Lovely Faculty of Technology and Sciences, Lovely Professional University, Phagwara, Punjab, India, 3 Department of Life Sciences, Presidency University, Kolkata, West Bengal, India, 4 Department of Medical Biochemistry, College of Medicine & Health Sciences, University of Gondar, Gondar, Ethiopia

* malikitrc@gmail.com (TM); abhijit.dbs@presiuniv.ac.in (AD)

## Abstract

### Introduction

To explore the ethnobiological wisdom of the tribals of three western districts of West Bengal, India against poisonous and non-poisonous bites and stings, a quantitative approach was adopted. These age-old yet unexplored knowledge can be utilized in finding lead-molecules against poisonous and non-poisonous animal-bites. Further, an evidence-based approach is needed to assess the venom-neutralization ability of plants by experimental studies.

### Materials and methods

During 2008–2009 and 2012–2017, 11 ethnomedicinal surveys were carried out to explore the use of medicinal flora and fauna via conducting open semi-structured interviews with 47 traditional healers (THs) or informants. The retrieved dataset was statistically evaluated using seven quantitative-indexes: use-value (UV), informants'-consensus-factor (ICF), fidelity-level (FL), relative-importance (RI), cultural importance-index (CI), index of agreement on remedies (IAR) and cultural agreement-index (CAI). Anti-phospholipaseA$_2$ (PLA$_2$) properties of selected plant extracts were also examined. In addition, the cytotoxicity and genotoxicity of the water extract of the plants showing high FL as well as significant PLA$_2$ inhibitory potential were investigated using *Allium cepa* root tip assay.

### Results

A total of 41 traditional-formulations (TFs) containing 40 plant species (of 39 genera from 28 families) and 3 animal species were prescribed by the THs. Fabaceae exhibited most

**Data Availability Statement:** All relevant data are within the manuscript and its Supporting Informations files.

**Funding:** The author(s) received no specific funding for this work.

**Competing interests:** The authors have declared that no competing interests exist.

number of medicinal plants. *Piper nigrum* (1.78) and *Apis cerana indica* and *Crossopriza lyoni* (both 0.21) exhibited the highest UV among the plants and the animals respectively. Stinging of centipede and dog/cat/hyena bite displayed highest ICF (1.00 each). Among the plants, the maximum RI (0.91) and CI (4.98) values were observed for *Aristolochia indica*. IAR (1.00) was recorded maximum for *Achyranthes aspera*, *Gloriosa superba*, *Lycopodium cernuum*, *Smilax zeylanica* and *Streblus asper*. Maximum CAI value was noted for *Piper nigrum* (5.5096). Among the animals, *Apis cerana indica* (0.31) and *Crossopriza lyoni* (1.52) displayed the highest RI and CI values respectively. *Crossopriza lyoni* (0.99) and *Apis cerana indica* (1.3871) exhibited maximum IAR and CAI values respectively. Plants showing higher FL exhibited higher anti-PLA$_2$ activity via selective inhibition of human-group PLA$_2$. In addition, *Allium cepa* root tip assay has indicated the safety and/or toxicity of the plant parts prescribed by the THs. Root water extracts of *Aristolochia indica* and *Gloriosa superba* exhibited significant genotoxicity and cytotoxicity.

## Conclusions

Three western districts of West Bengal is the natural abode for many tribal and non-tribal communities. A noteworthy correlation was established between the plants used against poisonous-bites and their anti-PLA$_2$ activity. A few plant parts used by the THs also exhibited high toxicity. Such alternative medical practices serve as the only option in these under-privileged and backward areas during medical-exigencies.

## Introduction

Snake envenomation is considered as a major problem worldwide especially in the tropical and subtropical countries including South East Asia [1–4]. There are more than 440,000 snake envenoming and 20,000 deaths each year [2]. Antivenom is considered as an effective treatment against snakebite; however snakebite-associated mortality remains significantly high due to unavailability, limited therapeutic efficacy and safety concerns of conventional antivenins [5]. India presents a very high number of snakebite incidences compared to other countries. Insufficient hospital-based reports indicated the total number of snakebite mortality to be ranging over 1,300 to 50,000 in India [6]. Besides snakebites, scorpion stings also cause severe venom-related injury [7]. Scorpion stings result in severe consequences in human which may lead to even mortality due to neurotoxins present in the venom [8]. Centipede bites have reportedly caused local pain, erythema and edema, nausea and vomiting, headache, lymphade-nopathy, rhabdomyolysis, myocardial infarction, hypotension etc. in humans due to myocardial toxic effects of the venom and anaphylaxis [9, 10]. Bee stings are also known to cause heart blocks, syncope and cardiac arrest [11]. Even the stings of the members of Hymenoptera such as bees, hornets, and wasps can be fatal [12]. Animal bites are reported as a serious global health concern and are responsible for almost 1–2% of all visits to the hospital emergency. Most of the animal bites are imposed by dogs (80–90%) and cats (5–15%) and the most common complication following animal bite is wound infection [by *Pasteurella multocida*, *Capno-cytophaga canimorsus*, *Eikenella corrodens* and Rhabdovirus (rabies only) etc.] which may lead to sepsis and death especially in the immunocompromised victims [13, 14].

There has always been a philosophical conflict between the mainstream Western medicine and the traditional and complementary medicine or TCM with their possible coexistence and

prevalence in the global context [15]. World Health Organization (WHO) has advocated herbal medicines as a valid alternative therapy against many human ailments. According to the WHO, almost 80% of the world's population rely on TCM for primary healthcare [16]. In India, other alternative medication strategies such as Ayurveda, Siddha, Unani, Tibbi and Homoeopathy have also been popularly used [17]. Purulia, one of the remote and backward districts of India is rich in aboriginals with their age old ethnomedicinal treatments. Remoteness from the nearby towns, unfavorable topography, sparse healthcare facilities and poverty have persuaded the aboriginals to use ethnomedicines especially during serious medical exigencies such as poisonous envenomations.

Phospholipase $A_2$ ($PLA_2$) enzymes, commonly present in the venoms of the snakes from the families Viperidae, Hydrophidae and Elaphidae are implicated to the venom-induced toxic effects [18–20]. Many natural compounds from plants, marine organisms, serum plasma etc. have been tested for their anti-$PLA_2$ properties on snake venoms and/or isolated toxins [20]. Higher $PLA_2$ inhibitory properties of the plant extracts indicate their possible snake venom neutralization ability. Earlier, *in-vitro*, *in-vivo* and *in-silico* studies performed on the anti-$PLA_2$ properties of the natural compounds indicated the tremendous potential of the compounds as novel inhibitors of ophidian toxins [21–23].

Standardized *Allium cepa* root tip test is commonly used as a fast and reliable method to assess cytotoxicity and genotoxicity of plant extracts, isolated compounds, synthetic derivatives, nano-materials, environmental pollutants, pesticides, industrial effluents etc. mostly on the basis of mitotic anomalies and chromosomal aberrations studies [24–28]. Medicinal plants, although used for their therapeutic properties, also exhibit dose dependent toxic effects. *Allium cepa* root tip meristematic cells assay easily determines dose dependent cytotoxicity and genotoxicity of medicinal plant extracts and plant-derived compounds [29–32].

The primary aim of the present study is to enumerate the indigenous use of phyto- and zoo-therapeutics prescribed by the traditional healers (THs) practicing in the three westernmost districts of West Bengal, India, against poisonous and non-poisonous animal bites and insect stings. The present work is also intended to elucidate the preparations and applications of traditional formulations (TFs) and the statistical interpretation of the retrieved ethnobiological information. In the present study, anti-$PLA_2$ properties of selected plant extracts were also evaluated against human group $PLA_2$ and porcine group $PLA_2$ to find out any selective inhibition of the said extracts on pro-inflammatory human group $PLA_2$ without or minimally inhibiting the porcine group digestive $PLA_2$. In addition, *Allium cepa* root tip meristematic cells were employed to assess the genotoxic and cytotoxic potential of selected antivenin plants showing higher fidelity level (FL) as well as superior $PLA_2$ inhibitory potential.

## Materials and methods

### Ethics statement

The permission (ethnomedicinal survey, sanctioned in 2012) was granted by the West Bengal Biodiversity Board [OM No. 040/3/K(Bio)−1/2012], Govt. of West Bengal, India. West Bengal Biodiversity Board is a Govt. of the State of West Bengal who issues necessary permission for ethnobiological surveys. Earlier, preliminary information on the area and the people were gathered by an approval granted by West Bengal Biodiversity Board (1.1.2008–22.5.2009) [Memo No. 5k(bio)-2/2007]. This Govt. body has approved this work which compiles with the guidelines. In addition, written and verbal consent were taken from the local people during the survey which also compiles to the ethical guidelines as provided by the aforesaid Govt. body.

## Selection of informants and data collection

During 2008–2009, initial surveys were conducted to explore the people and their practices in different villages of the three districts (Fig 1). Later, during 2012–2017, a total number of 11 ethnomedicinal surveys were conducted to explore the use of medicinal flora and fauna to treat poisonous and non-poisonous animal bites and insect stings. A total number of 47 THs (also known as the informants) were interviewed on the basis of their ethnomedicinal knowledge as well as their reputation and social acceptability as practitioners. A semi-structured questionnaire (S1 and S2 Figs) was supplied to each TH to explain the use (against particular ailments) and composition of each monoherbal or polyherbal ethnomedicinal preparations, their local names, plant/animal parts used, detection of disease symptoms, method of composing formulations, route of administration, addition of other ingredients etc. Before the start of the questionnaire session, prior consent was sought from the informants (S3 Fig). Besides, name, gender, age, aboriginal group etc. of the THs were also noted. Habit, habitat, taxonomic and identification features of the ethnobiologicals as well as date, time, season and site of collection were also documented. Herbarium sheets were prepared using medicinal plants preferably in their flowering phases and specific voucher numbers were assigned to each. Plants, plant parts and methods of preparations were also photographed for documentation. Literatures such as Prain (1903) [33] and Sanyal (1994) [34] were used to identify local flora and Tropicos plant database from Missouri Botanical Garden (www.tropicos.org) was consulted to verify names, synonyms and authors' citations. Economic Botany Data Collections Standards (EBDCS) proposed by Cook (1995) [35] was used to present the use categories mentioned by the informants (Table 1). Herbarium specimens with respective voucher numbers were preserved at the Department of Zoology, Sidho-Kanho-Birsha University, Purulia, West Bengal. Ethnozoologicals were initially documented during the field visits and were subsequently identified by the first author of this article. Animal samples were photographed and common samples were preserved for future reference.

## Analyses of ethnobiological data

**Use Value (UV).**   Use value (UV) is a quantitative analysis applied to enumerate the relative importance of local ethnobiologicals [36]. The equation is:

$$UV = \Sigma U/n$$

(where U denotes the number of citations per ethnobiological and n denotes the informant numbers who were interviewed for a given ethnobiological)

**Informant Consensus Factor (ICF).**   The informant consensus factor (ICF) was calculated to evaluate user variability of ethnobiologicals [37]. The ICF is estimated by the following equation:

$$ICF = n_{ur} - n_t/n_{ur} - 1$$

(where $n_{ur}$ is the used citation number in each category and $n_t$ is the number of ethnobiologicals reported)

**Fidelity Level (FL).**   The fidelity level (FL) is determined as the percentage of informants reporting the application of specific ethnobiological for similar purpose [38]. It was calculated as:

$$FL(\%) = N_p/N \times 100$$

(where $N_p$ is the informant numbers separately reporting an application of an ethnobiological

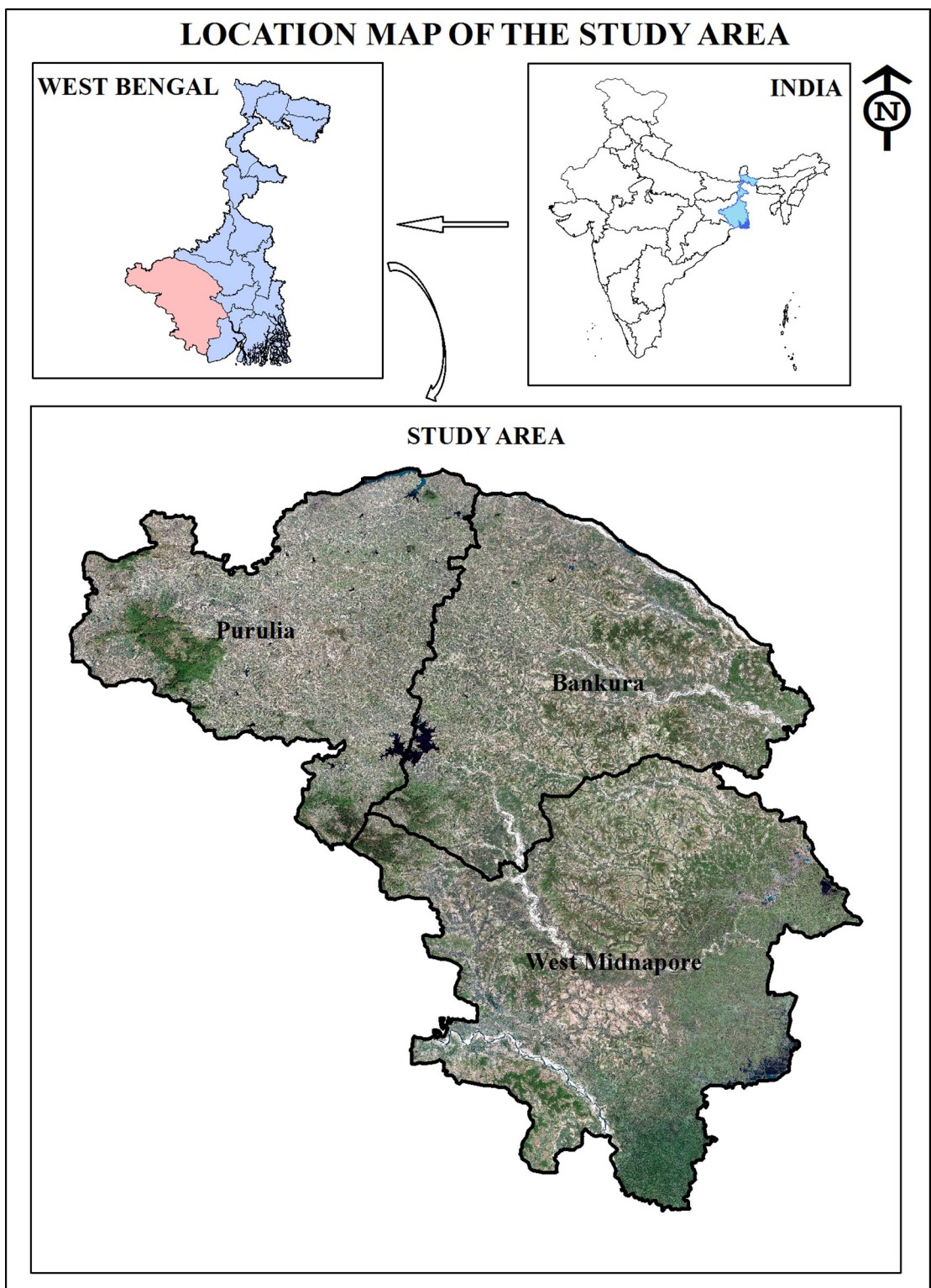

**Fig 1. Map of the study area (maps not to scale) (Map was created using the editor tools of the ArcGIS 10.3.1 software.).**

**Table 1. Traditional Formulations (TFs) used against poisonous and non-poisonous animal bites and insect stings.**

**Disease/ Disorder[1] 1: Scorpion sting (20090400)**

**symptoms: pain and burning sensation around stinging area and full organ**

| Traditional Formulations (TFs) | Composition[3] | Modes of preparation and methods of administration[2] |
|---|---|---|
| TF1 | *Bisalya karani* (*Barleria lupulina*) bark: 50 g | Bark is grinded to make a paste; the paste is used as ointment (1070000) on the wound. |
| TF2 | *Khakra change*/*Bandarlej* (*Setaria glauca*) roots: 10 g | Roots are grinded to make a paste; the paste is applied on the wound. |
| | *Tar genda* (*Tridax procumbens*) roots: 10 g | |
| | *Bhin Kambal* (*Premna herbacea*) roots: 10 g | |
| TF3 | *Rasun* (*Allium sativum*) bulbs: 5 in no. | Roots and *Rasun* bulbs are grinded to make a paste; the paste is applied on the wound with a pinch of salt. |
| | *Nirbis*i (*Cyperus kyllingia*) root: 20 g | |
| | Common salt (NaCl): a pinch | |
| TF4 | *Seowra* (*Streblus asper*) leaves: 4–5 in no. | Leaves are taken and rubbed by hand to extract juice and the juice is given on the stinging area. |
| TF5 | *Alkushi* (*Mucuna pruriens*) seeds: 2–3 in no. | A powder is made from the seeds; the powder is applied over the stinging areas. |
| TF6 | *Harin singha* (*Euphorbia tirucalli*) latex: 3–4 drops | Applied on the stinging area. |
| TF7 | *Gave verenda* (*Jatropha gossypifolia*) bark | Bark paste is directly applied on the affected area. |
| TF8 | Scorpion (*Buthus* sp.) | Internal material/ digestive system of scorpion by which the person was attacked is to rub on stinging area. |
| TF9 | *Ol* (*Amorphophallus paeoniifolius*) stem | The stem is heated over the flame and the affected area is covered with this heated stem. |
| TF10 | *Kend* (*Diospyros melanoxylon*) bark | Bark juice is applied on the stinging area. |
| TF11 | *Kalojira* (*Nigella sativa*) seeds | Grinded seeds are applied on the stinging area. |
| TF12 | *Liyaya ful*/ *Bishlanguli* (*Gloriosa superba*) roots | Fresh root is rubbed on the affected area. |
| TF13 | *Chatpati ghash* (*Ruellia tuberosa*) leaves: 10 g | Paste is applied on the affected area. |
| | *Tarnd mula* (*Premna herbacea*) roots: 10 g | |
| TF14 | *Narkol* (*Cocos nucifera*) oil: 5 to 10 ml | Both the ingredients are mixed well; the mixture is used as an ointment on the stinging area. |
| | Naphthalene: 1 ball | |
| TF15 | *Sada dhurba ghash* (*Cynodon dactylon*) leaves: 10 g | Both the ingredients are crushed to make a paste; the paste with *kalo saban* is applied on the stinging area. |
| | *Boka benri* (*Ipomoea carnea*) leaves: 10 g | |
| | *Kapor kachar kalo saban* (local black soap for washing clothes): 5 g | |

**Disease/ Disorder 2: Stinging of honey bee or wasp or hornet (20090200 and 20090300)**

**Symptoms: Swelling of affected area and severe burning sensation**

| | | |
|---|---|---|
| TF16 | *Hetal* (*Crateva adansonii*) bark | Bark of *Hetal* is grinded to make paste. Bark paste is applied on the wound. |
| TF17 | *Lal Tulsi* (*Ocimum sanctum*) leaves: 50 g | Leaf paste is smeared over the body to keep away bees. |
| TF18 | *Narkol* (*Cocos nucifera*) oil: 5 to 10 ml | Both the ingredients are mixed well. Mixture is used like ointment on stinging area. |
| | Naphthalene: 1 ball | |
| TF19 | Common salt (NaCl): ½ teaspoonful | Application of a lotion (1050000) prepared from the salt and the kerosene oil just after stinging is helpful to relieve pain. |
| | Kerosene oil: 1 teaspoonful | |
| TF20 | Petrol: 5–10 ml | After removing the sting, cotton is soaked with petrol and it is rubbed on the stinging area until the burning sensation is gone. Then juice of *Piyaj* is then applied on the affected area. |
| | Cotton/cotton cloth: 1 in no. | |
| | *Piyaj* (*Allium cepa*) bulb: 50 g | |
| TF21 | *Liyaya ful*/*Bishlanguli* (*Gloriosa superba*) roots | Fresh root is rubbed on the affected area. |
| TF22 | *Baichi* (*Flacourtia indica*) stem and bark: 10 g | A paste of stem and bark is applied on the wound. |
| TF23 | *Rangani*/*Shiyal kanta* (*Argemone mexicana*) seeds: 10 g | The seeds are crushed well to make a paste and the paste is applied on the stinging area. |

**Disease/ Disorder 3: Stinging of centipede**

| | | |
|---|---|---|
| TF24 | *Ol* (*Amorphophallus paeoniifolius*) stem | The stem is heated over the flame and the affected area is covered with this heated stem. |

(*Continued*)

**Table 1.**  (Continued)

| **Disease/ Disorder[1] 1: Scorpion sting (20090400)** | | |
|---|---|---|
| **symptoms: pain and burning sensation around stinging area and full organ** | | |
| **Traditional Formulations (TFs)** | **Composition[3]** | **Modes of preparation and methods of administration[2]** |
| **Disease/ Disorder 4: Dog/cat/hyena bite (12020000)** | | |
| TF25 | *Chitchiti* (*Achyranthes aspera*) roots: 20 g | Root paste is applied on the wound and molasses is taken orally. |
|  | Molasses: 20 g |  |
| TF26 | *Crossopriza lyoni* (Blackwall, 1867) (tailed cellar spiders): 1–2 in no. | Ingredients are properly mixed and taken orally to reduce the effect of the poison. |
|  | *Chanchh*i (The deposits at the bottom of a pot after the milk is boiled): 50 g |  |
| TF27 | *Tentul* (*Tamarindus indica*) tender twigs | Freshly taken twigs are heated over the flame and to cover the wound. |
| **Disease/ Disorder 5: Snake bite (20090600)** | | |
| TF28 | *Bisalya karani* (*Barleria lupulina*) bark: 50 g | Bark is grinded to make a paste. The paste is used as ointment at the wound. |
| TF29 | *Mrita sanjiboni* (*Lycopodium cernuum*) leaves: 50 g | All the ingredients are grinded to make a paste; the paste is applied on the wound. |
|  | *Asthi sancharini* (*Scindapsus officinalis*) leaves: 50 g |  |
|  | *Bisalya karani* (*Barleria lupulina*) leaves: 50 g |  |
| TF30 | *Chotopard* (*Cissampelos pareira*) roots: 10 g | Root is grinded to make a paste; the paste is fed to the victim to prevent the spread of the poison. |
| TF31 | *Bamun hati*/*Ram datum* (*Smilax zeylanica*) roots: 1 inch | Root of *Ram datum* and *Iswarmul* along with *Rabing* seeds are grinded to form a paste; the paste is applied on the wound. |
|  | *Iswarmul* (*Aristolochia indica*) roots: 1 inch | If patient becomes senseless, powder of dry leaves and roots of *Kanch mala* is given to the victim as snuff. |
|  | Black pepper or *Rabing* (*Piper nigrum*) seeds: 2½ in no. |  |
|  | *Kanch mala* (*Abrus precatorius*) dry leaves: 5 in no. |  |
|  | *Kanch mala* roots: 2 g |  |
| TF32 | *Dusatin lata* (*Gloriosa superba*) bulb | *Dusatin lata* is purified with bovine urine for seven days. The purified *Dusatin lata* is taken with dust of black pepper. |
|  | Black pepper or *Rabing* (*Piper nigrum*) seeds |  |
|  | Bovine urine |  |
| TF33 | White *Kuch fall*/*Kukurmala* (*Abrus precatorius*) roots: 2 ½ g | Roots of *Kuch fall* and black pepper are grinded together and taken orally. |
|  | Black pepper or *Rabing* (*Piper nigrum*) seeds: 10 in no. |  |
| TF34 | *Ishwarmul* (*Aristolochia indica*) roots: 1½ inch | Paste is made and given to the patient orally. |
|  | *Rangani* (*Solanum surattense*) roots: 1½ inch |  |
|  | *Bagh nakhi* (*Martynia annua*) roots: 1½ inch |  |
|  | *Piyal* (*Buchanania lanzan*) bark: 1½ inch |  |
|  | Black pepper or *Golki*/*Rabing* (*Piper nigrum*) seeds: 12 in no. |  |
| TF35 | *Ishwarmul* (*Aristolochia indica*) bark and roots: 5 g | Juice of bark and root is drunk twice daily. |
| TF36 | *Barachadar* (*Rauvolfia canescens*) roots: 10 g | 10 ml of root juice is prescribed orally and is also applied on the wound. |
| TF37 | *Anantamul*/*Analsing* (*Hemidesmus indicus*) roots: 10 g | Root paste is applied on the wound. |
|  | *Apis cerana indica* Fabricius (Indian honeybee) fresh honey from hives: 50 ml |  |

(*Continued*)

**Table 1.** (*Continued*)

| **Disease/ Disorder[1] 1: Scorpion sting (20090400)** | | |
|---|---|---|
| **symptoms: pain and burning sensation around stinging area and full organ** | | |
| **Traditional Formulations (TFs)** | **Composition[3]** | **Modes of preparation and methods of administration[2]** |
| TF38 | *Ishermul/Bhedi-Janete* (*Aristolochia indica*) roots | Root paste with a paste of ten peppers is given as antidote. |
| | Black pepper or *Golki/Rabing* (*Piper nigrum*) seeds: 10 in no. | |
| TF39 | *Chotopar/Chotkipar/Tijumala* (*Cissampelos pareira*) roots | Root paste with a paste of ten peppers is given as antidote. |
| | Black pepper or *Golki/ Rabing* (*Piper nigrum*) seeds: 10 in no. | |
| TF40 | *Mahadevjata/Ishwarjata* (*Uraria picta*) leaves | Leaf paste is given as an antidote |
| TF 41 | *Swet akanda* (*Calotropis gigantea*) leaves | A paste is prepared with all the ingredients and is offered to the patient to drink. |
| | *Bishaynandi* (*Glossogyne bidens*) whole plant | |
| | *Krishnajata* (*Nardostachys jatamansi*) roots | |

[1, 2] The numbers used in parenthesis after the diseases/disorders and methods of administration are according to the recommendations given by Cook, 1995 as Economic botany data collection standard (EBDCS), plant parts, body parts and processes, disorders/effects, medicinal applications and non-vertebrate organisms (Master lists of states for Level 3 descriptors) (Economic Botany Data Standard; https://www.kew.org/tdwguses/rptMasterListMain.htm).

[3]In composition, in no. used after the numbers stands for in number i.e. the number of that plant part used.

to treat a specific disease whereas N is the total number of informants citing the use of an ethnobiological to treat any specific ailment)

**Relative Importance (RI).** The relative importance (RI) of an ethnobiological depicts the most used ethnobiologicals with the most notable number of ethnomedicinal applications [39, 40]. The RI is represented by the equation:

$$RI = PP + AC$$

(where PP is the total number of curative properties ascribed to an ethnobiological divided by the highest number of similar activities ascribed to the widely used ethnobiologicals. AC represents the number of ailment categories reported to be cured by a specific ethnobiological divided by the highest number of ailment categories treated by the most prolifically used ethnobiologicals).

**Cultural Importance index (CI).** The cultural importance index (CI) is calculated by the total number of informants citing the use of each ethnobiological [41]. The CI is represented by:

$$CIs = \sum_{u=u1}^{uNC} \sum_{i=i1}^{iN} URui/N$$

(where UR represents the use report, N represents the gross number of informants whereas from $u_1$ to $u_{NC}$ represent UR of each use-category [41, 42]).

**Index of Agreement on Remedies (IAR).** The index of agreement on remedies (IAR) depicts the significance of each ethnobotanicals [43]. IAR is calculated as:

$$IAR = n_r - n_a/n_{r-1}$$

(where $n_r$ is the summation of the reports on the use of an ethnobiological and $n_a$ is the number of ailment categories against which the ethnobiological is administered [44]).

**Cultural Agreement Index (CAI).** The cultural agreement index (CAI) [45] is calculated as:

$$CAI = CII \text{ X } IARs$$

(where CII represents the cultural importance index and IAR represents the index of agreement on remedies)

## Enzymes and reagents

Human group $PLA_2$ and porcine group $PLA_2$ were used as $PLA_2$. Lecithin (egg yolk phosphatidylcholine), red phenol, sodium taurodeoxycholate (NaTDC) and ethylmethanesulfonate (EMS) were procured from Sigma, India. A UV-VIS spectrophotometer was used for the measurements using the visible spectra.

## $PLA_2$ inhibitory activity

$PLA_2$ inhibitory activity of medicinal plants were tested only with the plants showing high FL. A protocol by De Aranjo and Radvany (1987) was followed to assess the $PLA_2$ inhibitory activity of the plant extracts [46]. In this study, the inhibitory effects of the water extracts were assessed on the pro-inflammatory human group $PLA_2$. As a control set, porcine group PLA2 was used. The substrate was composed of egg yolk phosphatidylcholine or lecithin (3.5 mM) in a mixture containing NaTDC (3 mM), NaCl (100 mM), $CaCl_2$ (10 mM) and red phenol (0.055 mM) employed as colorimetric indicator dissolved in $H_2O$ (100 ml). The pH of the reaction mixture was calibrated to 7.6. The two $PLA_2$ were solubilized in acetonitrile (10%) at 0.02 and 0.002 µg/µl concentrations respectively. Each plant extract (10 ml) was incubated with $PLA_2$ solution (10 µl) at room temperature for 20 min. Following this, $PLA_2$ substrate (1 ml) was added, and following hydrolysis in next 5 min, the OD was read at 558 nm. The inhibition percentage was determined by comparing with the control (absence of plant extract) and the $IC_{50}$ values were obtained from the curve.

## *Allium cepa* root tip meristem genotoxicity test

A protocol by Aşkin Çelik and Aslantürk (2010) with minor modifications was employed to assess the genotoxicity and cytotoxicity of medicinal plants with high FL as well as superior human-group $PLA_2$ inhibitory properties [47]. Tap water was used as negative control whereas ethyl methanesulfonate (EMS, $2 \text{ X } 10^{-2}$ M) was employed as positive control. After 24 h exposure to the control sets and the aqueous extract (2.5, 5 and 10 mg/ml) of the plants, *Allium cepa* root tips were cut from the bulbs to fix them in ethanol: glacial acetic acid: ethanol (1:3 v/v) overnight at 4 ˚C. Following fixation, the roots were put in 70% (v/v) aqueous ethanol and were stored in a fridge. The root tips were then hydrolyzed using hydrochloric acid (HCl) (1N) for 3 min and were subsequently squashed on microscope slides with 45% acetic acid after staining with aceto-orcein [2% (w/v)]. Slides were visualized under a Carl Zeiss compound microscope with 40X10 magnification (n = 5/set). The cytotoxic and genotoxic features visualized were i. mitotic-index (MI) which is the ratio of the total number of mitotically dividing cells and the total number of cells present in a microscopic field expressed in percentage; ii. micronuclei (MNC) formation during interphase cells per 1000 cells (‰MNC) and iii. chromosomal anomalies *viz.* chromosomal fragments, anaphase bridges, multipolarity, laggard chromosomes and ghost cells.

## Results

### Demographic profile of the THs

Among the 47 THs interviewed, 41 were men and 6 were women aged between 32 to 85 years. Thirty three male THs were having primary or parallel professions like agriculture, animal husbandry or other services whereas the women were all housewives. Remaining eight male THs practiced traditional medicine as their primary profession. District-wise, among 47 THs, 33 represented Purulia and 8 and 6 THs represented Bankura and West Midnapore districts respectively. Most THs have been practicing traditional medicine for years which they acquired from their ancestry.

### Traditional therapeutics using ethnobiologicals

A total number of 41 TFs were prescribed by the THs against poisonous and non-poisonous animal bites and insect stings (Table 1). They have reported 40 plant species (of 39 genera from 28 families) and 3 animal species (of 3 genera from 3 families) as direct ingredients in TFs against 5 poisonous and non-poisonous animal bites and insect stings. Among plant families, Fabaceae exhibited most number of medicinal plants (4 species) (Fig 2). A total number of 3 animal species were reported (Table 1), parts of which were used as direct ingredients or as additives in TFs. Salt, soap, kerosene oil, petrol, molasses, black pepper, milk, bovine urine and honey were added as additives, as taste enhancers and/or to enhance the efficacy of formulations applied topically. The plant and animal species with their local names, families and part (s) used are presented in Table 2 indicating their use in respective TFs and against the particular type of bite. Plant voucher numbers and habit types are also tabulated in Table 2. Herbs (20 records) exhibited the most common plant habit (Fig 3) whereas roots (21 uses) were reported as the most commonly used plant part (Fig 4).

### Preparations and applications of TFs

Paste or ointment (23 uses) was recorded as the most used form of preparations (Fig 5) and topical administration was the most common mode of application (Fig 6). Among the animal parts, whole body, digestive system and honey were used. In most of the cases, plant and/or

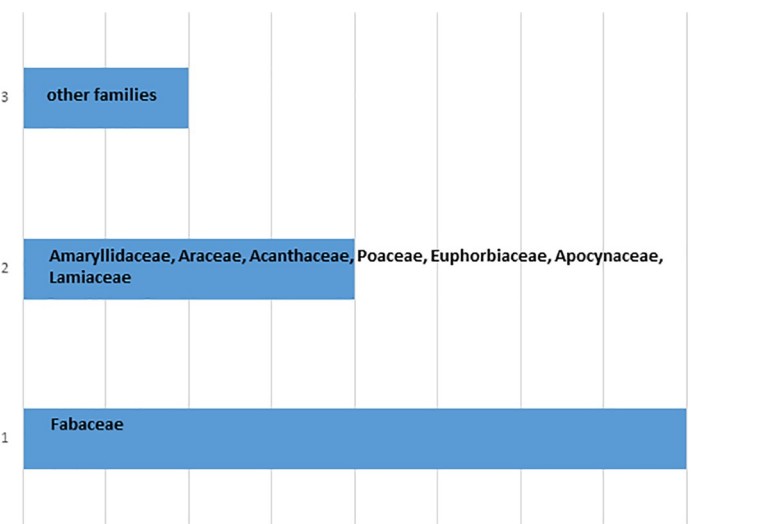

**Fig 2. Distribution of plant families.**

**Table 2. Details of plants and animals as ingredients in Traditional Formulations (TFs) against poisonous and non-poisonous animal bites and insect stings.**

| Botanical/Zoological binomials | Voucher No.[1] | Family | Vernacular names | Habit | Part(s) used[2] | Used in TFs[3] | Ailment/ disorder treated[4] | UV | RI | CI | IAR | CAI |
|---|---|---|---|---|---|---|---|---|---|---|---|---|
| *Abrus precatorius* L. | PB001 | Fabaceae | *Kuch fall/ Kukurmala/Kanch Mala* | climber | root (12040000) | 31, 33 | 5 | 0.22 | 0.31 | 1.43 | 0.98 | 1.4014 |
| *Achyranthes aspera* L. | PB002 | Amaranthaceae | *Chitchiti* | herb | root | 25 | 4 | 0.23 | 0.34 | 1.54 | 1.00 | 1.54 |
| *Argemone mexicana* L. | PB036 | Papavaveraceae | *Rangani/ Shiyal kanta* | herb | seed (11000000) | 23 | 2 | 0.24 | 0.39 | 1.59 | 0.99 | 1.5741 |
| *Allium cepa* L. | PB003 | Amaryllidaceae | *Piyaj* | herb | bulb (12020000) | 20 | 2 | 0.27 | 0.37 | 1.57 | 0.98 | 1.5386 |
| *Allium sativum* L. | PB004 | Amaryllidaceae | *Rasun* | herb | bulb | 3 | 1 | 0.31 | 0.37 | 2.66 | 0.98 | 2.6068 |
| *Amorphophallus paeoniifolius* (Dennst.) Nicolson | PB005 | Araceae | *Ol* | herb | stem (6000000) | 9, 24 | 1, 3 | 0.48 | 0.55 | 3.71 | 0.99 | 3.6729 |
| *Apis cerana indica* Fabricus (8031100) | - | Apidae | *Moumachhi* | - | honey | 37 | 5 | 0.21 | 0.31 | 1.43 | 0.97 | 1.3871 |
| *Aristolochia indica* L. | PB006 | Aristolochiaceae | *Iswarmul/ Ishermul/Bhedi-Janete* | herb | root, bark (7000000) | 31, 34, 35, 38 | 5 | 1.59 | 0.91 | 4.98 | 0.99 | 4.9302 |
| *Barleria lupulina* Lindl. | PB007 | Acanthaceae | *Bisalya karani* | herb | bark | 1, 28, 29 | 1, 5 | 1.38 | 0.89 | 3.87 | 0.98 | 3.7926 |
| *Buchanania lanzan* Spreng. | PB008 | Anacardiaceae | *Piyal* | tree | bark | 34 | 5 | 0.20 | 0.26 | 1.55 | 0.98 | 1.519 |
| *Buthus* sp. Leach, 1815 (8010100) | - | Buthidae | *Kankra bichha* | - | digestive system (6000000) | 8 | 1 | 0.19 | 0.25 | 1.44 | 0.97 | 1.3968 |
| *Cissampelos pareira* L. | PB009 | Menispermaceae | *Chokipar/ Tijumala/ Chotopard* | climber | root | 30, 39 | 5 | 1.11 | 0.79 | 3.65 | 0.98 | 3.577 |
| *Cocos nucifera* L. | PB010 | Arecaceae | *Narcol* | tree | oil | 14, 18 | 1, 2 | 0.98 | 0.77 | 3.60 | 0.98 | 3.528 |
| *Crateva adansonii* DC. | PB011 | Capparaceae | *Hetal* | tree | bark | 16 | 2 | 0.20 | 0.30 | 1.49 | 0.97 | 1.4453 |
| *Crossopriza lyoni* (Blackwall, 1867) (8010200) | - | Pholcidae | *Makadsa* | - | whole body (11040000) | 26 | 4 | 0.21 | 0.28 | 1.52 | 0.99 | 1.5048 |
| *Cynodon dactylon* (L.) Pers. | PB037 | Poaceae | *Sada dhurba ghash* | herb | leaf (8000000) | 15 | 1 | 0.20 | 0.29 | 1.49 | 0.97 | 1.4453 |
| *Cyperus kyllingia* Endl. | PB012 | Cyperaceae | *Nirbishi* | herb | root | 3 | 1 | 0.21 | 0.30 | 1.52 | 0.99 | 1.5048 |
| *Diospyros melanoxylon* Roxb. | PB013 | Ebenaceae | *Kend* | tree | bark | 10 | 1 | 0.23 | 0.29 | 1.57 | 0.97 | 1.5229 |
| *Euphorbia tirucalli* L. | PB014 | Euphorbiaceae | *Harin singha* | tree | latex (13020000) | 6 | 1 | 0.24 | 0.32 | 1.57 | 0.99 | 1.5543 |
| *Flacourtia indica* (Burm. f.) Merr. | PB015 | Salicaceae | *Baichi* | tree | stem, bark | 22 | 2 | 0.22 | 0.32 | 1.53 | 0.98 | 1.4994 |
| *Gloriosa superba* L. | PB016 | Colchicaceae | *Liyaya ful/ Bishlanguli/ Dusatin lata* | herb | root | 12, 21, 32 | 1, 2, 5 | 1.25 | 0.88 | 3.43 | 1.00 | 3.43 |
| *Glossogyne bidens* (Retz.) Alston | PB039 | Asteraceae | *Bishaynandi* | herb | whole plant (2000000) | 41 | 5 | 0.21 | 0.28 | 1.47 | 0.99 | 1.4553 |
| *Hemidesmus indicus* (L.) R. Br. ex Schult. | PB017 | Apocynaceae | *Anantamul/ Analsing* | herb | root | 37 | 5 | 0.29 | 0.37 | 1.59 | 0.98 | 1.5582 |
| *Ipomoea carnea* Jacq. | PB038 | Convolvulaceae | *Boka benri* | shrub | leaf | 15 | 1 | 0.28 | 0.38 | 1.55 | 0.98 | 1.519 |
| *Jatropha gossypifolia* L. | PB018 | Euphorbiaceae | *Gave verenda* | shrub | bark | 7 | 1 | 0.22 | 0.33 | 1.49 | 0.97 | 1.4453 |
| *Lycopodium cernuum* L. | PB019 | Lycopodiaceae | *Mrita sanjibani* | herb | leaf | 29 | 5 | 0.23 | 0.33 | 1.50 | 1.00 | 1.5 |
| *Martynia annua* L. | PB020 | Martyniaceae | *Bagh nakhi* | herb | root | 34 | 5 | 0.23 | 0.29 | 1.51 | 0.98 | 1.4798 |

*(Continued)*

**Table 2.** (Continued)

| Botanical/Zoological binomials | Voucher No.[1] | Family | Vernacular names | Habit | Part(s) used[2] | Used in TFs[3] | Ailment/ disorder treated[4] | UV | RI | CI | IAR | CAI |
|---|---|---|---|---|---|---|---|---|---|---|---|---|
| *Mucuna pruriens* Scop. | PB021 | Fabaceae | *Alkushi* | shrub | seed | 5 | 1 | 0.31 | 0.28 | 1.77 | 0.97 | 1.7169 |
| *Nardostachys jatamansi* (D. Don) DC. | PB040 | Caprifoliaceae | *Krishnajata* | herb | root | 41 | 5 | 0.21 | 0.29 | 1.41 | 0.98 | 1.3818 |
| *Nigella sativa* L. | PB022 | Ranunculaceae | *Kalojira* | herb | seed | 11 | 1 | 0.29 | 0.35 | 1.75 | 0.99 | 1.7325 |
| *Ocimum sanctum* L. | PB023 | Lamiaceae | *Lal tulsi* | herb | leaf | 17 | 2 | 0.30 | 0.34 | 1.86 | 0.99 | 1.8414 |
| *Piper nigrum* L. | PB024 | Piperaceae | *Rabing/Golki* | vine | seed | 31, 32, 33, 34, 38, 39 | 5 | 1.78 | 0.80 | 5.68 | 0.97 | 5.5096 |
| *Premna herbacea* Roxb. | PB025 | Lamiaceae | *Tarnd mula* | herb | root | 2 | 1 | 0.25 | 0.33 | 1.49 | 0.98 | 1.4602 |
| *Rauvolfia canescens* L. | PB026 | Apocynaceae | *Barachadar* | shrub | root | 36 | 5 | 0.25 | 0.31 | 1.54 | 0.98 | 1.5092 |
| *Ruella tuberosa* L. | PB027 | Acanthaceae | *Chatpati ghash* | herb | leaf | 13 | 1 | 0.24 | 0.30 | 1.65 | 0.99 | 1.6335 |
| *Scindapsus officinalis* (Roxb.) Schott | PB028 | Araceae | *Asthi sancharini* | climber | leaf | 29 | 5 | 0.20 | 0.29 | 1.49 | 0.97 | 1.4453 |
| *Setaria glauca* (L.) P. Beauv. | PB029 | Poaceae | *Khakra change/ Bandarlej* | herb | root | 2 | 1 | 0.21 | 0.29 | 1.50 | 0.98 | 1.47 |
| *Smilax zeylanica* L. | PB030 | Smilacaceae | *Bamunhati/ Ramdatun* | shrub | root | 31 | 5 | 0.25 | 0.29 | 1.55 | 1.00 | 1.55 |
| *Solanum surattense* Burm. f. | PB031 | Solanaceae | *Rangani* | herb | root | 34 | 5 | 0.19 | 0.25 | 1.46 | 0.99 | 1.4454 |
| *Streblus asper* Lour. | PB032 | Moraceae | *Seowra* | tree | leaf | 4 | 1 | 0.22 | 0.31 | 1.49 | 1.00 | 1.49 |
| *Tamarindus indica* L. | PB033 | Fabaceae | *Tentul* | tree | twig | 27 | 4 | 0.27 | 0.31 | 1.60 | 0.97 | 1.552 |
| *Tridax procumbens* L. | PB034 | Asteraceae | *Tar genda* | herb | root | 2 | 1 | 0.26 | 0.32 | 1.58 | 0.98 | 1.5484 |
| *Uraria picta* (Jacq.) Desv. | PB035 | Fabaceae | *Mahadevjata/ Ishwarjata* | herb | leaf | 40 | 5 | 0.22 | 0.32 | 1.49 | 0.99 | 1.4751 |

[1]Voucher no. indicates the herbarium sheet number deposited for future reference.

[2]The numbers used in parenthesis after the parts used are according to the recommendations given by Cook, 1995 as Economic botany data collection standard (EBDCS), plant parts, body parts and processes, disorders/effects, medicinal applications and non-vertebrate organisms (Master lists of states for Level 3 descriptors) (Economic Botany Data Standard; https://www.kew.org/tdwguses/rptMasterListMain.htm).

[3]Used in TFs indicates the plants used in traditional formulations given in Table 1. UV: use value; RI: Relative importance; CI: Cultural importance index; IAR: Index of agreement on remedies; CAI: cultural agreement index.

[4]The numbers use in this column are in accordance to the numbers used for different disease/ disorders in Table 1.

animal parts were ground to powder to make an ointment and were applied topically. In a few cases, the plant parts were heated over the flame and the affected area was covered with this heated part. TF19 was found to be applied as a lotion prepared from the salt and the kerosene oil and was applied on the stinging area to relieve pain just after an insect-sting. Most of the preparations were used either orally or topically; however, TF36 was used both orally as well as topically on the wound. In another interesting study, in TF8, the internal material/ digestive system of scorpion (*Buthus* sp.) by which the person was stung was rubbed on the stinging area. In TF26, *Crossopriza lyoni* (tailed cellar spiders) was used with plant part and milk and was taken orally to reduce the effect of the poison.

## Quantitative ethnobiology: UV, ICF, FL, RI, CI, IAR and CAI

*Piper nigrum* (1.78) and *Apis cerana indica* and *Crossopriza lyoni* (both 0.21) exhibited the highest UV among the plants and the animals respectively (Table 2). Stinging of centipede and dog/cat/hyena bite displayed highest ICF (ICF = 1.00 each) (Table 3). The ethnobotanicals/

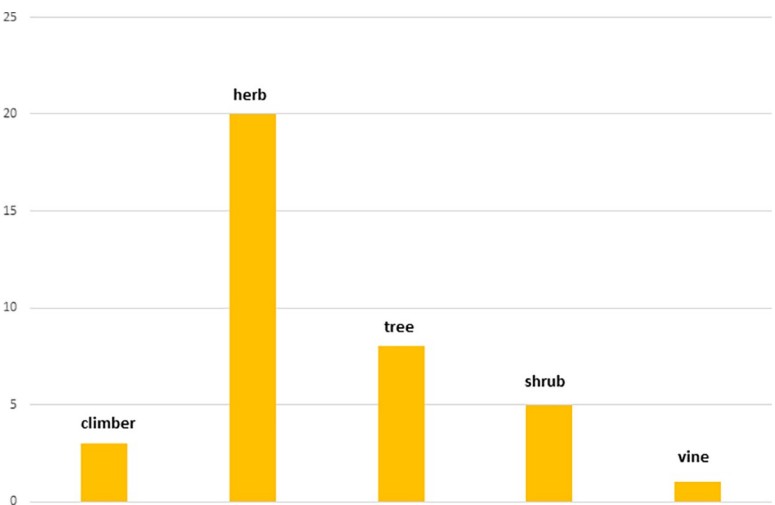

**Fig 3. Distribution of plant habit types.**

ethnozoologicals exhibiting high (70–100%), moderate (50–70%) and low (<50%) FL are presented in Table 4.

Other quantitative indices *viz*. RI, CI, IAR and CAI were also calculated for the plants and animals used as ethnomedicine (Table 2). Among the plants, maximum RI value (0.91) as well as CI value (4.98) were observed for *Aristolochia indica*. IAR which was calculated on the basis of importance of each species, was recorded maximum (1.00) for 5 plant species such as *Achyranthes aspera*, *Gloriosa superba*, *Lycopodium cernuum*, *Smilax zeylanica* and *Streblus asper*. Maximum CAI value was noted for the plant species *Piper nigrum* (5.5096). Among the animals, *Apis cerana indica* (0.31) displayed the highest RI value and *Crossopriza lyoni* (1.52) exhibited maximum CI value. *Crossopriza lyoni* also exhibited maximum IAR value (0.99) whereas *Apis cerana indica* (1.3871) was recorded for the maximum CAI value.

## Use of ethnobotanicals: Toxicity aspects and conservation status

The use of the whole body of tailed cellar spider *Crossopriza lyoni* and the digestive system of the scorpion *Buthus* sp. may be implicated to possible toxicity and adverse effects in the

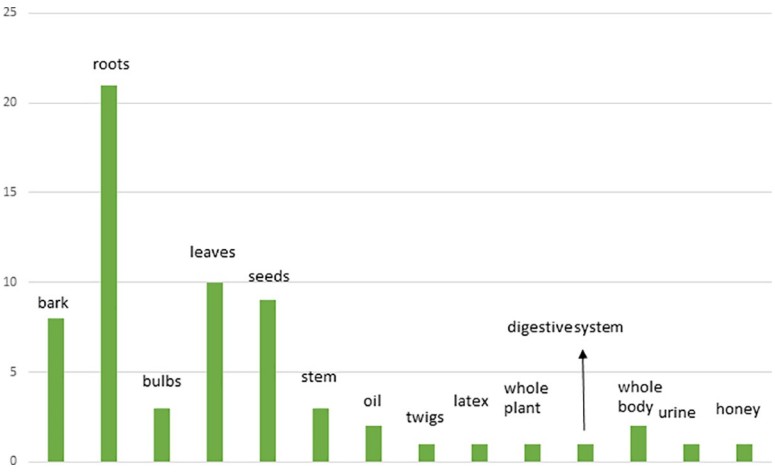

**Fig 4. Distribution of plant and animal parts used by the THs.**

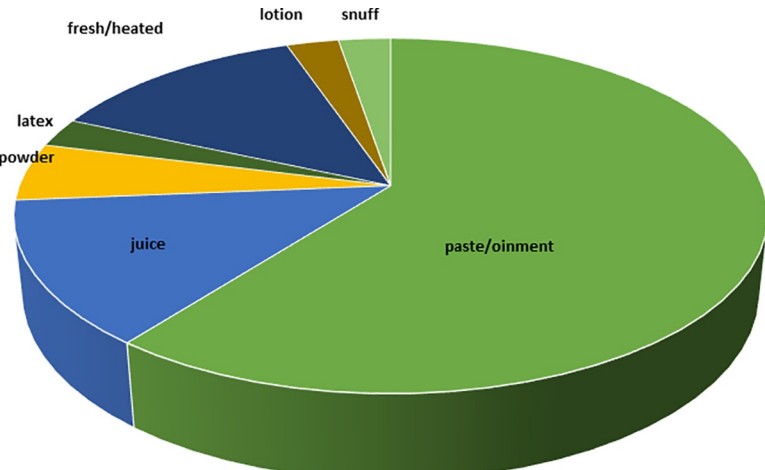

**Fig 5. Distribution of drug preparations used by the THs.**

recipients of oral or topical mode of administration. However, no such toxicity was reported by the THs. One of the prime conservation strategies adopted by the tribal people was to protect the plants by worshipping them in sacred groves. The authors found that plants such as *Tamarindus indica*, *Cissampelos pareira*, *Streblus asper*, *Calotropis gigantea*, *Abrus precatorius*, *Ocimum sanctum*, *Achyranthes aspera* and *Aristolochia indica* etc. were conserved in sacred groves.

## Determination of PLA$_2$ inhibition

The objective here was to find out plant extracts showing selective inhibition against the pro-inflammatory human group PLA$_2$ without or minimally suppressing the porcine group digestive PLA$_2$. Initially, three extracts [water, methanol and chloroform-ethanol (1:1)] exhibited promising outcome regarding PLA$_2$ inhibition. However, only water extract was analyzed further since most of the ethnomedicines were prepared using water. The water extract of *Aristolochia indica* roots demonstrated the most significant inhibition of the enzyme human PLA$_2$ with an IC$_{50}$ value of 0.73 mg/ml. Following *Aristolochia indica*, water extracts of other plant

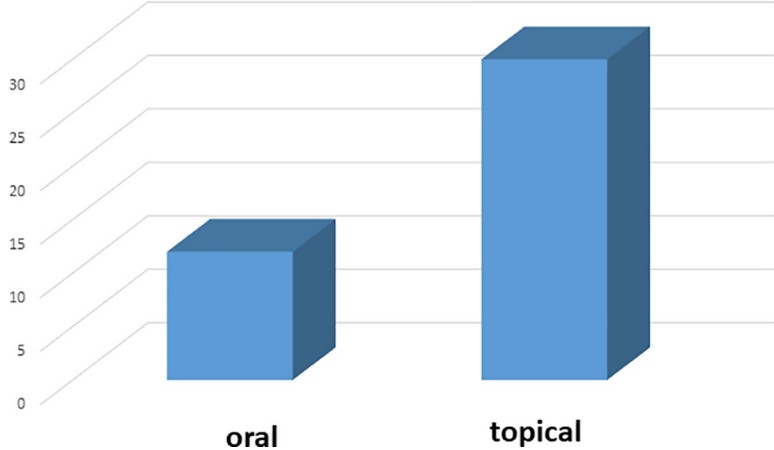

**Fig 6. Distribution of modes of drug administration prescribed by the THs.**

**Table 3. Sting/bite category and corresponding Informant Consensus Factor (ICF) depicted from the interviews with the THs.**

| Sting/bite category | $n_{ur}$ = number of use citations in each category | $n_t$ = number of species used against a particular ailment by all informants | Informant consensus factor (ICF) $n_{ur} - n_t / n_{ur} - 1$ |
|---|---|---|---|
| Scorpion sting | 28 | 22 | 0.22 |
| Stinging of honey bee or wasp or hornet | 16 | 9 | 0.47 |
| Stinging of centipede | 4 | 1 | 1 |
| Dog/cat/hyena bite | 7 | 1 | 1 |
| Snake bite | 30 | 25 | 0.17 |

parts showing promising $PLA_2$ inhibition were *Mucuna pruriens* seeds ($IC_{50}$ value = 0.79 mg/ml), *Allium cepa* bulbs, *Gloriosa superba* roots (both with $IC_{50}$ value = 0.81 mg/ml), *Hemidesmus indicus* roots ($IC_{50}$ value = 0.83 mg/ml), *Nardostachys jatamansi* roots ($IC_{50}$ value = 0.88 mg/ml), *Rauvolfia canescens* roots ($IC_{50}$ value = 0.90 mg/ml) and *Piper nigrum* seeds ($IC_{50}$ value = 0.97 mg/ml) (Table 5). These findings indicated a selective inhibition of the extracts against the two $PLA_2$.

## Assessment of cytotoxicity and genotoxicity by *Allium cepa* root tip assay

Aqueous extracts of *Allium cepa* bulb, *Hemidesmus indicus* root, *Nardostachys jatamansi* root and *Piper nigrum* seed did not affect MI significantly since they demonstrated almost similar results as shown by the negative control. *Mucuna pruriens* seed and *Rauvolfia canescens* root water extracts exhibited slight inhibition of MI whereas the roots of *Aristolochia indica* and *Gloriosa superba* exhibited significant inhibition of MI especially at higher concentrations. Although the positive control EMS ($2 \times 10^{-2}$ M) showed maximum inhibition of MI (%) (1.58), *Aristolochia indica* root extracts at 10 mg/ml concentration showed potent antimitotic activity in *Allium cepa* root meristems with an MI (%) of 1.77. In addition, with increasing concentration of the water extracts of the used plant parts, a gradual decline in the MI was noted which was statistically comparable with both the controls (Fig 7). A few plant part extracts, at higher concentration caused chromosomal aberrations such as chromosome breaks, anaphase bridges, multipolarity, laggard chromosomes and ghost cells. However, the chromosomal aberrations were most profound in root tips treated with the positive control EMS. Genotoxicity of the root extracts was also recorded in the interphase MNC formation which was expressed as per 1000 cells (‰MNC). The root water extracts of *Aristolochia indica* and *Gloriosa superba* exhibited maximum genotoxic and cytotoxic potential which was comparable to the EMS. MNC formation was also found to be the highest when root tip cells were treated with the root water extracts of *Aristolochia indica* and *Gloriosa superba* (Table 6).

## Discussion

### Study area and the aboriginals

The three western districts of the state of West Bengal, India are Purulia, Bankura and West Midnapore. This area is an extension of the Chota Nagpur Plateau (22˚-25˚ 30'N and 83˚47'-87˚ 50'E). Santhalis, Oraons, Mundas, Bhumijs, Birhors, Gonds, Kharias, Mal Pahariyas and Hos are the major tribal groups residing in these three districts. Austro-Asiatic and Dravidian languages are common among the tribal people. Natural tropical forests are prevalent in the westernmost district i.e. Purulia; however the forests have been constantly decreasing due to human exploitation. In this area, forests have supported the livelihood of its inhabitants via providing food, fodder, clothing, building materials, timber and so on. Undulated topography

**Table 4. Fidelity Level (FL) of the ethnobiologicals used depicted from the interviews with the THs.**

| Disease/ailment category | FL (%) category | Most favored plant/animal used against particular bite/sting |
|---|---|---|
| Scorpion sting | High FL (70–100%) | *Allium sativum* |
| | | *Barleria lupulina* |
| | | *Cynodon dactylon* |
| | | *Cyperus kyllingia* |
| | | *Gloriosa superba* |
| | | *Ipomoea carnea* |
| | | *Mucuna pruriens* |
| | | *Nigella sativa* |
| | | *Premna herbacea* |
| | | *Setaria glauca* |
| | Moderate FL (50–70%) | *Diospyros melanoxylon* |
| | | *Euphorbia tirucalli* |
| | | *Jatropha gossypifolia* |
| | | *Streblus asper* |
| | Low FL (<50%) | *Amorphophallus paeoniifolius* |
| | | *Buthus sp.* |
| | | *Cocos nucifera* |
| | | *Ruellia tuberosa* |
| Stinging of honey bee or wasp or hornet | High FL (70–100%) | *Allium cepa* |
| | | *Cocos nucifera* |
| | | *Ocimum sanctum* |
| | Moderate FL (50–70%) | *Argemone Mexicana* |
| | | *Flacourtia indica* |
| | | *Gloriosa superba* |
| | Low FL (<50%) | *Crateva adansonii* |
| Stinging of centipede | Moderate FL (50–70%) | *Amorphophallus paeoniifolius* |
| Dog/cat/hyena bite | Moderate FL (50–70%) | *Achyranthes aspera* |
| | | *Crossopriza lyoni* |
| | | *Tamarindus indica* |
| Snake bite | High FL (70–100%) | *Abrus precatorius* |
| | | *Apis cerana indica* |
| | | *Aristolochia indica* |
| | | *Cissampelos pareira* |
| | | *Gloriosa superba* |
| | | *Hemidesmus indicus* |
| | | *Piper nigrum* |
| | | *Rauvolfia canescens* |
| | | *Smilax zeylanica* |
| | Moderate FL (50–70%) | *Barleria lupulina* |
| | | *Buchanania lanzan* |
| | | *Glossogyne bidens* |
| | | *Martynia annua* |
| | | *Nardostachys jatamansi* |
| | | *Scindapsus officinalis* |
| | Low FL (<50%) | *Calotropis gigantea* |
| | | *Lycopodium cernuum* |
| | | *Solanum surattense* |
| | | *Uraria picta* |

FL: fidelity level.

**Table 5. PLA$_2$ inhibitory activities of the water extracts of the ethnobotanicals with high FL.**

| Ethnobotanicals and parts | IC$_{50}$ values (mg/ml) on human group PLA$_2$ | IC$_{50}$ values (mg/ml) on porcine group PLA$_2$ | Inhibition specificity (IC$_{50}$ porcine group PLA$_2$/IC$_{50}$ human group PLA$_2$) |
|---|---|---|---|
| *Abrus precatorius* L. root | 1.76 | >5 | >2.84 |
| *Achyranthes aspera* L. root | 1.89 | >5 | >2.64 |
| *Argemone mexicana* L. seed | 1.67 | >5 | >2.99 |
| *Allium cepa* L. bulb | 0.81 | 3.2 | 3.95 |
| *Allium sativum* L. bulb | 1.98 | >5 | >2.52 |
| *Amorphophallus paeoniifolius* (Dennst.) Nicolson stem | 2.20 | >5 | >2.27 |
| *Aristolochia indica* L. root | 0.73 | 3.1 | 4.24 |
| *Barleria lupulina* Lindl. bark | 2.34 | >5 | >2.13 |
| *Buchanania lanzan* Spreng. bark | 3.34 | >5 | >1.49 |
| *Cissampelos pareira* L. root | 3.76 | >5 | >1.33 |
| *Cocos nucifera* L. oil | 3.67 | >5 | >1.36 |
| *Crateva adansonii* DC. bark | 2.89 | >5 | >1.73 |
| *Cynodon dactylon* (L.) Pers. leaf | 2.67 | >5 | >1.87 |
| *Cyperus kyllingia* Endl. root | 3.45 | >5 | >1.45 |
| *Diospyros melanoxylon* Roxb. bark | 4.54 | >5 | >1.10 |
| *Euphorbia tirucalli* L. latex | 3.09 | >5 | >1.62 |
| *Flacourtia indica* (Burm. f.) Merr. stem | 4.12 | >5 | >1.21 |
| *Gloriosa superba* L. root | 0.81 | 3.3 | 4.07 |
| *Glossogyne bidens* (Retz.) Alston whole plant | 2.98 | >5 | 1.67 |
| *Hemidesmus indicus* (L.) R. Br. ex Schult. root | 0.83 | 3.4 | 4.09 |
| *Ipomoea carnea* Jacq. leaf | 3.67 | >5 | >1.36 |
| *Jatropha gossypifolia* L. bark | 3.09 | >5 | >1.61 |
| *Lycopodium cernuum* L. leaf | 4.86 | >5 | >1.03 |
| *Martynia annua* L. root | 4.71 | >5 | >1.06 |
| *Mucuna pruriens* Scop. seed | 0.79 | 3.2 | 4.05 |
| *Nardostachys jatamansi* (D. Don) DC. root | 0.88 | 2.9 | 3.29 |
| *Nigella sativa* L. seed | 3.22 | >5 | >1.55 |
| *Ocimum sanctum* L. leaf | 3.33 | >5 | >1.50 |
| *Piper nigrum* L. seed | 0.97 | 3.6 | 3.71 |
| *Premna herbacea* Roxb. root | 4.60 | >5 | >1.08 |
| *Rauvolfia canescens* L. root | 0.90 | >5 | >5.55 |
| *Ruella tuberosa* L. leaf | 4.23 | >5 | >1.18 |
| *Scindapsus officinalis* (Roxb.) Schott leaf | 3.49 | >5 | >1.43 |
| *Setaria glauca* (L.) P. Beauv. root | 3.32 | >5 | >1.50 |
| *Smilax zeylanica* L. | 3.48 | >5 | >1.43 |
| *Solanum surattense* Burm. f. root | 4.04 | >5 | >1.23 |
| *Streblus asper* Lour. root | 3.19 | >5 | >1.56 |
| *Tamarindus indica* L. leaf | 4.04 | >5 | >1.23 |
| *Tridax procumbens* L. twig | 4.54 | >5 | >1.10 |
| *Uraria picta* (Jacq.) Desv. leaf | 2.78 | >5 | >1.79 |

IC$_{50}$: The half maximal inhibitory concentration; PLA$_2$: phospholipases A2.

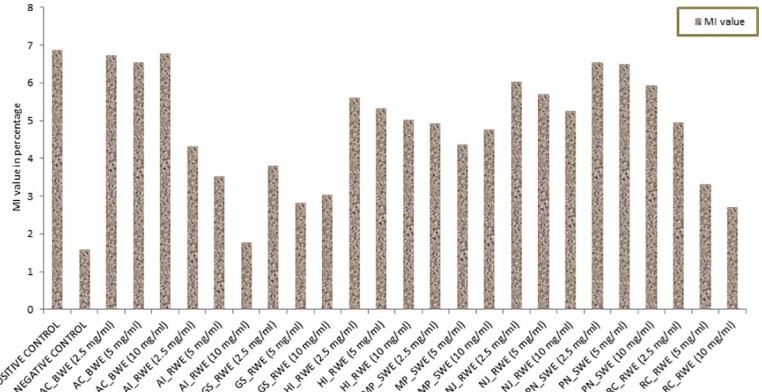

**Fig 7. Mitotic Index (MI) in the root meristematic cells of *Allium cepa* in control and treatment concentrations of water extract of ethnobotanicals with high Fidelity Level (FL).**

and extreme climate also counted for their overdependence on forest resources instead of opting for conventional agricultural practices. However the two districts Bankura and West Midnapore have cultivable lands and the presence of tribal groups is significantly lower in the two

**Table 6. Chromosomal aberrations and micronuclei formation in the root meristematic cells of *Allium cepa* in control and treatment concentrations of water extract of ethnobotanicals showing high FL and high PLA$_2$ inhibition.**

| Treatment groups | Concentrations | Chromosomal fragments ±SD | Multipolarity ±SD | Anaphase bridge ±SD | Laggard chromosome ±SD | Ghost cells ±SD | MNC (‰) ±SD |
|---|---|---|---|---|---|---|---|
| negative control (tap water) | - | - | - | - | - | - | - |
| positive control (EMS) | $2 \times 10^{-2}$ M | - | 9.61±4.49 | 4.02±3.56 | 3.39±2.45 | 13.90±3.32 | 0.48±0.19 |
| *Allium cepa* L. bulb, water extract | 2.5 mg/ml | - | - | - | - | - | - |
| | 5 mg/ml | - | - | - | - | - | - |
| | 10 mg/ml | - | - | - | - | - | - |
| *Aristolochia indica* L. root, water extract | 2.5 mg/ml | 6.65±1.19 | 7.85±2.21 | 1.91±2.22 | 2.46±3.33 | 10.59±2.87 | 0.54±0.23 |
| | 5 mg/ml | 5.76±0.46 | 7.04±1.09 | 2.08±1.87 | 1.50±2.09 | 11.38±3.90 | 0.49±0.22 |
| | 10 mg/ml | 6.09±0.44 | 6.28±2.24 | 2.23±1.89 | 1.70±3.45 | 11.10±4.05 | 0.44±0.45 |
| *Gloriosa superba* L. root, water extract | 2.5 mg/ml | 4.66±0.37 | 6.84±1.76 | 2.66±0.71 | 1.02±0.67 | 12.76±3.67 | 0.39±0.17 |
| | 5 mg/ml | 5.54±1.45 | 5.50±0.89 | 1.89±0.98 | 2.81±0.98 | 12.06±4.56 | 0.59±0.14 |
| | 10 mg/ml | 4.81±0.67 | 6.93±0.69 | 1.94±0.91 | 1.19±0.20 | 10.90±2.68 | 0.60±0.09 |
| *Hemidesmus indicus* (L.) R. Br. ex Schult. root, water extract | 2.5 mg/ml | 1.67±0.76 | - | - | - | 1.27±0.22 | - |
| | 5 mg/ml | 2.02±0.34 | - | - | - | 2.32±1.48 | - |
| | 10 mg/ml | 1.88±1.73 | - | - | 0.56±2.831.75 | 2.01±1.09 | - |
| *Mucuna pruriens* Scop. seed, water extract | 2.5 mg/ml | 2.02±1.77 | 4.77±0.73 | 0.67±2.56 | 0.98±1.79 | 7.70±2.67 | - |
| | 5 mg/ml | 1.72±0.80 | 5.07±0.66 | 1.04±1.29 | 0.68±1.20 | 8.93±2.90 | 0.12±0.08 |
| | 10 mg/ml | 1.90±0.47 | 4.17±1.39 | 1.87±1.62 | 1.10±1.11 | 8.88±1.96 | 0.11±0.17 |
| *Nardostachys jatamansi* (D. Don) DC. root, water extract | 2.5 mg/ml | 1.05±0.33 | 2.21±1.98 | - | - | 2.76±2.69 | - |
| | 5 mg/ml | 0.66±0.21 | 1.65±1.54 | - | - | 2.90±2.34 | - |
| | 10 mg/ml | 1.14±0.20 | 2.94±2.02 | - | - | 1.98±2.93 | - |
| *Piper nigrum* L. seed, water extract | 2.5 mg/ml | - | - | - | - | - | - |
| | 5 mg/ml | - | - | - | - | - | - |
| | 10 mg/ml | - | - | - | - | - | - |
| *Rauvolfia canescens* L. root, water extract | 2.5 mg/ml | 3.45±0.22 | 5.09±1.45 | 1.56±1.98 | 1.92±2.34 | 9.09±2.82 | 0.29±0.18 |
| | 5 mg/ml | 4.98±0.29 | 4.96±0.46 | 2.20±2.74 | 2.02±2.67 | 11.10±2.78 | 0.32±0.23 |
| | 10 mg/ml | 4.02±33 | 5.01±0.48 | 1.85±2.89 | 2.22±3.03 | 10.63±3.22 | 0.17±0.25 |

districts compared to Purulia. Over the past few years, irrigation and social forestry have improved the agricultural practices and productivity as well as forest coverage in the aforesaid areas. Once, Purulia was mentioned as the one of the 250 most backward districts (out of 640) of India, for which it received fund from the Backward Regions Grant und Program (BRGF) [48]. Non-availability of mainstream medication and remoteness of hospitals contribute to their over-reliance on age-old traditional medicines prescribed by THs. However, recent technological and communicational advancement alongside medical and backward development facilities have lifted the standard of living of the tribal populations. This elevation in social and economic conditions in the tribal populations subsequently declined the use of traditional medication that also led to the indifference and ignorance among the younger generations in taking up traditional healing as professions. Hence, the present documentation is also a timely representation of rapidly vanishing medicinal folklore of the local ethnic groups.

## Traditional therapeutics using ethnobiologicals

Earlier, a number of ethnobiological surveys were conducted globally to explore the use of plants and animals against poisonous bites in northwest Colombia [49], Ethiopia [50], Hainan Island, China [51] and so on. Moreover, plants used against snakebites in Santarém, western Pará, Brazil were validated scientifically against hemorrhagic activity caused by *Bothrops jararaca* venom [52]. Ethnobotanical use of antivenin plants used in Uganda were also evidenced by pharmacological analysis [53]. Neutralization of lethal, enzymatic and hemorrhagic effects of *Bothrops atrox* venom by medicinal plants from Colombia was documented in a series of studies [49, 54, 55].

Some plants *viz. Hemidesmus indicus* [56], *Gloriosa superba* [57], *Tamarindus indica* [58], *Aristolochia indica* [59–61] etc. depicted in the present study have been investigated previously *in vitro* and/or *in vivo* for evaluating the anti-venom properties of their extracts and phytochemicals. The present study also reveals a few animals or animal products used in the folkloric medicine in polyherbal formulations active against poisonous or non-poisonous bites. The authors have also noted the inseparable existence of magical beliefs and prescribed medicine. For example, in TF8, the scorpion envenomation victim is given the internal material/ digestive system of the same scorpion by which the person was stung.

In India, up to one million snake bites are recorded in a single year of which as many as 50,000 are recorded as initial death. In fact, roughly the number of people died of snakebites in India, is almost equal to the total deaths due to snakebite in the rest of the world. And yet most of these deaths could have been prevented if necessary medical care was taken. Till date, the snake anti-venom is the only treatment for poisonous snakebites, which is produced from the snake venom itself. Most of the snake bites are either dry bites or from non-poisonous snakes. Still death occurs. There are many instances where patients die only because of fear and psychological shock. THs provide at least first aid and mental support to both patient and his/her families, where there is no other option [62, 63].

## Preparations and applications of TFs

Mode of administration of TFs against poisonous and non-poisonous bites seemed to have played a crucial role since few of the plant or animal materials would have caused toxicity if applied orally instead of topical application. Authors have noted that most of these medicines were applied on the bitten or stinging area as paste or ointment or lotion, which was obviously meant to relieve the pain instantly or to alleviate symptoms of poisonous bites at some later stages. Among the TFs, only TF26, TF30, TF32, TF33, TF35 and TF41 were taken orally. Rest of the TFs were applied topically. In case of TF31, if patient becomes senseless, powder of dry

leaves and roots of *Kanch mala* is given to the victim as snuff. As a prophylactic measure, *Lal Tulsi* (*Ocimum sanctum*) leaves were smeared over the body to keep away bees. In TF25, *Chit-chiti* (*Achyranthes aspera*) root paste and molasses were prescribed topically and orally respectively against dog/cat/hyena bite.

## Quantitative ethnobiology

Relative importance of the plants and animals used in the ethnomedicines is indicated by UV and among the plants *Piper nigrum* exhibited maximum UV (1.78) because of its widespread use in TFs to attenuate bitter taste of the formulations. *Aristolochia indica* (1.59), *Barleria lupulina* (1.38) and *Gloriosa superba* (1.25) were recorded among the other botanicals with high UVs which reflects their widespread acceptance as ethnomedicine. Similarly, *Aristolochia indica* (0.91), *Barleria lupulina* (0.89) and *Gloriosa superba* (0.88) also exhibited high RI values. The highest CI and CAI values were displayed by *Piper nigrum* which were 5.68 and 5.95 respectively. Highest IAR (1.00) was exhibited by four plants such as *Achyranthes asper*a, *Gloriosa superba*, *Lycopodium cernuum* and *Streblus asper*. In comparison to the plants, very less number of animals were present in the TFs which is reflected in their low UVs. High/moderate/low FL is displayed by various plant or animal species since they have been differentially accepted by the THs in their formulations.

## Use of plants and animals as drugs: Toxicity and conservation aspects

Despite being used since time immemorial, use of traditional medicine must consider safety and toxicity issues. Animal samples such as digestive system of scorpion, whole body of spider and animal excreta such as bovine urine may possess toxicity and pathogens causing adverse effects in the recipients. Topical application of naphthalene, kerosene and petrol is also needed to be assessed for toxicity and side effects. THs have reported *Aristolochia indica* as a potent anti-venom herb. However, it has also been reported to cause aristolochic acid nephropathy (AAN) in different parts of the globe which has led to the discontinuation of the drug in different herbal products containing aristolochic acid [64]. Among the ethnozoologicals, *Apis cerana indica*, *Buthus* sp. and *Crossopriza lyoni* have not yet been assessed by the International Union for Conservation of Nature and Natural Resources (IUCN) Red List [source: The IUCN Red List of Threatened Species[TM] (http://www.iucnredlist.org/search))].

## PLA$_2$ inhibitory activity of the medicinal plants

Interestingly, most of the plants (*Aristolochia indica*, *Mucuna pruriens*, *Allium cepa*, *Gloriosa superba*, *Hemidesmus indicus*, *Nardostachys jatamansi*, *Rauvolfia canescens* and *Piper nigrum*) showing high FL (70–100%) exhibited high PLA$_2$ inhibitory activity against human group PLA$_2$ without or minimally inhibiting the digestive porcine group PLA$_2$. Similar observations were noted while elucidating the anti-PLA$_2$ activity of *Aloe vera* leaf skin extracts. Earlier, selective inhibition of pro-inflammatory PLA$_2$ group IIA was attributed to the catechin tannins present in the water extract of *Aloe vera*. It was further suggested that the compounds present in the extracts exhibiting anti-PLA$_2$ properties were different than those displaying antioxidant properties [65]. High PLA$_2$ inhibitory potential of the plants indicates their possible effectiveness against snake venom. In exploration of alternative antivenin, especially in rural India where lack of conventional antivenin against poisonous snakebites causes a number of mortality and morbidity every year. Available literature also supports the traditional use of *Aristolochia indica*, *Gloriosa superba*, *Hemidesmus indicus* [66] and *Nardostachys jatamansi* [67] against snakebite. Snake venom neutralization ability of *Hemidesmus indicus* [55], *Aristolochia indica* [58] and *Gloriosa superba* [68] has also been assessed via *in vitro* and *in vivo*

studies. A number of phytochemicals such as 2-hydroxy-4-methoxy benzoic acid and lupeol acetate from *Hemidesmus indicus* [69, 70] and aristolochic acid from *Aristolochia indica* [59, 60] also exhibited potent antivenin as well as anti-PLA$_2$ activities.

Aristolochic acid [8-methoxy-6-nitrophenanthro(3,4-*d*)-1,3-dioxole-5-carboxylic acid], an uncompetitive inhibitor with a K$_i$ of $9.9 \times 10^{-4}$ M (with phosphatidylcholine as substrate), was found to interact with the major basic PLA$_2$ from *Vipera russelli* venom. Administration of aristolochic acid inhibited edema-inducing activity of *Vipera russelli* PLA$_2$. Suppression of edema-induction by aristolochic acid was manifested when it reached the site; however, it did not aid in recovery. Aristolochic acid also failed to suppress other pathological properties of PLA$_2$ [60]. In another study, aristolochic acid inhibited human synovial fluid (HSF)-PLA$_2$, porcine pancreatic PLA$_2$, *Naja naja* PLA$_2$, and human platelet derived PLA$_2$ dose dependently with sensitivity of these PLA$_2$s to aristolochic acid varied significantly: HSF-PLA2> *Naja naja* PLA2> human platelet PLA$_2$> porcine pancreatic PLA2. In addition, it was indicated that inhibition of HSF-PLA$_2$ was possibly mediated via direct interaction with the enzyme [70]. The compound 2-hydroxy-4-methoxy benzoic acid from *Hemidesmus indicus* exhibited adjuvant efficacy and antiserum potentiation in rabbits immunized with *Vipera russellii* venom demonstrating potent venom neutralization ability (lethal and hemorrhage) [69]. *Hemidesmus indicus* root extract-derived lupeol acetate remarkably neutralized *Daboia russellii* venom-induced edema, haemorrhage, defibrinogenation, PLA$_2$ activity and lethality in male albino mice. Furthermore, *Naja kaouthia* venom-induced neurotoxicity, cardiotoxicity, respiratory modulations and lethality in the animals were also neutralized by the compound. In addition, venom-induced alterations in super oxide dismutase (SOD) activity and lipid peroxidation were also antagonized by lupeol acetate [70]. Therefore, ethnobiological use of a few medicinal plants reported in the present study is being supported by scientific literature describing their *in vitro* and *in vivo* efficacy as potent antivenin.

## Genotoxic and cytotoxic effects of plant extracts on *Allium cepa* root tip meristems

The plant parts used by the THs were extracted in water and the roots of *Aristolochia indica* and *Gloriosa superba* demonstrated significant antimitotic and genotoxic potential in *Allim cepa* root tip assay. Therefore, oral administration of such extracts are discouraged also keeping in mind the worldwide occurrence of aristolochic acid nephropathy (AAN) due to the consumption of *Aristolochia* preparations in traditional medicine and in health supplements [71]. Earlier, *Allium cepa* root tip meristems were used to assess the genotoxic and cytotoxic effects of *Aristolochia birostris* water and alcoholic extracts [72]. *Hemidesmus indicus* root extract was evaluated in cultured lymphocytes for its genotoxic and antigenotoxic effects [73]. Species of *Gloriosa* were also evaluated for their antimitotic effects on onion roots [74]. The present study evaluated the mutagenic and genotoxic properties of the plant extracts in a dose dependent manner and further suggested to take precautions before using few plant extracts in human.

## Conclusions

The three western districts of the state of West Bengal are the natural dwelling place for many indigenous communities surviving the climatic and economic hardships and exercising their time tested medical practices mostly based upon the uses of ethnobiologicals. These traditional and alternative treatments serve as the only option in these underprivileged and geographically remote areas during medical-exigencies like snake envenomation. The present study depicts a quantitative ethnobiological analysis among the aboriginals from the area against poisonous

and non-poisonous animal bites and insect stings. However, the efficacy of the antivenin eth-nobiologicals are needed to be validated scientifically. In addition, bioactivity guided isolation of phyto-constituents may lead to the templates for synthesis of novel antivenins. In addition, plants with higher FL displayed superior anti-PLA$_2$ properties via selective inhibition of human-group PLA$_2$. In addition, *Allium cepa* root tip assay revealed significant genotoxic and cytotoxic properties of some plant extracts. Therefore, concomitant studies on toxicology and safety of the plant extracts are also needed for safe efficacious application of botanical-derived antivenin.

## Supporting information

**S1 Fig. Interview data sheet.**
(DOCX)

**S2 Fig. Specimen copy of an interview data sheet.**
(DOCX)

**S3 Fig. A consent letter in Bengali language provided by one of the informants.**
(DOCX)

**S1 Graphical abstract.**
(PPTX)

## Acknowledgments

Local people of the study area are thankfully acknowledged for sharing ethnomedicinal infor-mation. We would also like to thank Mr Mrinal Mondal, Assistant Professor, Department of Geography, Sidho-Kanho-Birsha University, Purulia for constructing the map of the study area.

## Author Contributions

**Conceptualization:** Abhijit Dey, Tabarak Malik.

**Investigation:** Biplob Kumar Modak, Partha Gorai, Tabarak Malik.

**Methodology:** Biplob Kumar Modak, Partha Gorai, Abhijit Dey.

**Resources:** Biplob Kumar Modak, Partha Gorai.

**Supervision:** Devendra Kumar Pandey.

**Validation:** Devendra Kumar Pandey.

**Writing – original draft:** Biplob Kumar Modak, Abhijit Dey.

**Writing – review & editing:** Abhijit Dey, Tabarak Malik.

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
