## [Decision Letter · Decision Letter 0]

14 Jul 2020

PONE-D-20-17909

An evidenced based efficacy and safety assessment of the antivenom ethnobiologicals used by the tribals of three westernmost districts of West Bengal, India: Anti-phospholipase A2 and genotoxic effects

PLOS ONE

Dear Dr. Malik,

Thank you for submitting your manuscript to PLOS ONE. After careful consideration, we feel that it has merit but does not fully meet PLOS ONE’s publication criteria as it currently stands. Therefore, we invite you to submit a revised version of the manuscript that addresses the points raised during the review process.

The detailed reports of the reviewers are enclosed above. All questions deserve attention, especially the improvement of English language use and the discussion section's reformulation.

We look forward to receiving your revised manuscript.

Kind regards,

Benito Soto-Blanco, DVM, MSc, PhD

Academic Editor

PLOS ONE

Journal Requirements:

2. To ensure that you obtained ethics approval before your study began, please upload the ethics approval document issued by West Bengal Biodiversity Board approval (1.1.2008-22.5.2009) [Memo No. 5k(bio)-2/2007].

3. We note that [Figure(s) 1] in your submission contain [map/satellite] images which may be copyrighted. All PLOS content is published under the Creative Commons Attribution License (CC BY 4.0), which means that the manuscript, images, and Supporting Information files will be freely available online, and any third party is permitted to access, download, copy, distribute, and use these materials in any way, even commercially, with proper attribution. For these reasons, we cannot publish previously copyrighted maps or satellite images created using proprietary data, such as Google software (Google Maps, Street View, and Earth). For more information, see our copyright guidelines: http://journals.plos.org/plosone/s/licenses-and-copyright.

1.    You may seek permission from the original copyright holder of Figure(s) [1] to publish the content specifically under the CC BY 4.0 license. 

Reviewers' comments:

Reviewer's Responses to Questions

**Comments to the Author**

1. Is the manuscript technically sound, and do the data support the conclusions?

Reviewer #1: Yes

Reviewer #2: Partly

2. Has the statistical analysis been performed appropriately and rigorously? 

Reviewer #1: Yes

Reviewer #2: N/A

3. Have the authors made all data underlying the findings in their manuscript fully available?

Reviewer #1: Yes

Reviewer #2: Yes

4. Is the manuscript presented in an intelligible fashion and written in standard English?

Reviewer #1: Yes

Reviewer #2: No

5. Review Comments to the Author

Reviewer #1: The article “An evidenced based efficacy and safety assessment of the antivenom ethnobiologicals used by the tribals of three westernmost districts of West Bengal, India: Anti-phospholipase A2 and genotoxic effects.” presents a survey conducted with traditional healers from a specific region in India, aiming to catalogue traditional formulations used as treatments against animal bites and stings and assess their efficacy and safety using experimental assays with the reported ethnobotanicals. It is an interesting work that addresses an important matter because not only the traditional knowledge can drive further investigations on the pharmacological potential of the natural products, leading to new drugs, but it is also important to characterize possible toxic effects of such formulations in order to protect the population making use of them. Considering the neglected issue of envenoming and the scarcity of available treatments, and the fact that the populations living in poor and isolated districts are mostly affected by these problems, the presented article has merit.

I would recommend this work for publication in PLOSOne, but it must be submitted to a major revision, in order to attend the journal’s quality standards. First of all, the manuscript is to long. Information should be presented in a more concise and direct language, highlighting the main findings. For this, English language, although intelligible, must also be revised. Other issues are listed below:

Title: the work title mentions “antivenom ethnobiologicals”, but dog/cat/hyena bites, which do not contain venom, are listed as ailments for the traditional formulations. Whether the title should be modified or these particular conditions must be taken out from the results.

The interviewees are mentioned as “traditional healers” (TH) and “traditional medicine man” (TMM) along the text. This can cause confusion, so only one of the terms should be chosen.

Material and Methods: in the “Study area and aboriginals” session, the authors make narrative description that, although interesting, should be better placed in the discussion session. The Material and Methods should deliminate the study subject in more direct form.

The period of time when the survey was conducted differs between the ‘Abstract’ (2008-2009 and 2010-2017) and ‘Materials and Methods’. Which one is correct?

Table 1 would be better placed in the ‘Results session’.

In the Results session, it would be interesting to have the demographic profile of the interviewees (sex/age/profession/district).

The second session in ‘Results’ is named ‘Preparations, application and dose-dependence of TFs’, but there is no mention of the dose-dependence in the results.

Table 2 should me made horizontally, to increase readability and should be placed in the ‘Quantitative ethnobiology’ session. Also, in this table, what does “voucher number’ stands for? And the numbers that appear after the description of ‘parts used’? This information should be removed or explained in the table’s legend. Information on abbreviations is also lacking in table 5.

As a suggestion, data presented in table 6 would be more clearly visualized as a Graph.

Table 7 would also be better read horizontally.

The Discussion session, unlike the other session, is rather succinct. I believe the other parts of the manuscript could be more concise and Discussion should be further explored, comparing the found information with other similar articles, performed in other regions.

In line 419, authors state that ‘Most of the snake bites are either dry bites or from non-Poisonous snakes. Still death occurs. There are many instances where patients die only because of fear and psychological shock.’ This seemed odd to me. Are there any references to support this statement?

In line 494 of the ‘Conclusion’, authors claim that ‘Various conservation strategies have also been described as adopted by the indigenous people in order to use plant resources sustainably.’. I did not find the description of these strategies throughout the text.

Reviewer #2: A broad review regarding the English language, punctuation and spacing should be made.

Several references are mentioned incorrectly during the text, such as seen on lines 102, 105 and 109, for example.

During introduction, authors stress that PLA2 enzymes are responsible for the inflammatory effects of snake bites, and that plant extracts detain some sort of PLA2 neutralizing capacity. However, activity of PLA2 of both human and porcine origin are also evaluated. These subgroups must be more clearly determined in methodology and results and discussion.

Authors should classify the different Authors also claim to evaluate genotoxicity and cytotoxicity, and that should be clear as an aim of the present study as well. The aim of the study is not clearly outlined by the authors.

On Table 1, TF4 is explained as masticated by hand, a term with dubious meaning.

Table 2 is impossible to read and contains some of the main results from the experiment that should be clearly available to the reader.

Table 7 also needs formatting.

Discussion is very superficial, mainly replicating results obtained. It should be rewritten, stressing what each plant extract detains from a biochemical or pharmacological level that corroborates with a lesser or more prominent effect regarding PLA2 neutralization.

Since no aim of the study was properly established, conclusion is also in need of revision.

6. PLOS authors have the option to publish the peer review history of their article (what does this mean?). If published, this will include your full peer review and any attached files.

Reviewer #1: No

Reviewer #2: No

---

## [Author Response · Author response to Decision Letter 0]

24 Aug 2020

To ensure that you obtained ethics approval before your study began, please upload the ethics approval document issued by West Bengal Biodiversity Board approval (1.1.2008-22.5.2009) [Memo No. 5k(bio)-2/2007].

Please upload the completed Content Permission Form or other proof of granted permissions as an "Other" file with your submission. Thank you. The desired documents are attached. The document of project [Memo No. 5k(bio)-2/2007] carried out is approved by the Principal, Acchuram Memorial College, under Sidho Kanho Birsha University, the then institute of the first author. The same document also confirms the other projects carried out in this regard. The second document from West Bengal Biodiversity Board, a Govt. Body also confirms the same project [Memo No. 5k(bio)-2/2007] on “Preparation of people’s biodiversity register of Jhalda-Darda Gram Panchayat” carried out by the first author. This was issued by Sr. Research Officer, West Bengal Biodiversity Board. Authors having all these documents in original and can exhibit if needed. Permission Form or other proof of granted permissions as an "Other" file being submitted.

We note that [Figure(s) 1] in your submission contain [map/satellite] images which may be copyrighted. We require you to either (1) present written permission from the copyright holder to publish these figures specifically under the CC BY 4.0 license, or (2) remove the figures from your submission: Map was created using the editor tools of the ArcGIS 10.3.1 software. It is mentioned now in the caption of Fig. 1.

Reviewer #1: 

The article “An evidenced based efficacy and safety assessment of the antivenom ethnobiologicals used by the tribals of three westernmost districts of West Bengal, India: Anti-phospholipase A2 and genotoxic effects.” presents a survey conducted with traditional healers from a specific region in India, aiming to catalogue traditional formulations used as treatments against animal bites and stings and assess their efficacy and safety using experimental assays with the reported ethnobotanicals. It is an interesting work that addresses an important matter because not only the traditional knowledge can drive further investigations on the pharmacological potential of the natural products, leading to new drugs, but it is also important to characterize possible toxic effects of such formulations in order to protect the population making use of them. Considering the neglected issue of envenoming and the scarcity of available treatments, and the fact that the populations living in poor and isolated districts are mostly affected by these problems, the presented article has merit.

I would recommend this work for publication in PLOSOne, but it must be submitted to a major revision, in order to attend the journal’s quality standards. First of all, the manuscript is to long. Information should be presented in a more concise and direct language, highlighting the main findings. For this, English language, although intelligible, must also be revised. Thank you for your kind comments. Now after the revision, we have tried to use concise and direct language which has also been revised extensively.

Other issues are listed below:

Title: the work title mentions “antivenom ethnobiologicals”, but dog/cat/hyena bites, which do not contain venom, are listed as ailments for the traditional formulations. Whether the title should be modified or these particular conditions must be taken out from the results. Thank you for your kind comments. It is corrected as suggested by you.

The interviewees are mentioned as “traditional healers” (TH) and “traditional medicine man” (TMM) along the text. This can cause confusion, so only one of the terms should be chosen. Thank you for your kind comments. It is corrected as suggested by you.

Material and Methods: in the “Study area and aboriginals” session, the authors make narrative description that, although interesting, should be better placed in the discussion session. The Material and Methods should deliminate the study subject in more direct form.

 Thank you for your kind comments. It is corrected as suggested by you.

The period of time when the survey was conducted differs between the ‘Abstract’ (2008-2009 and 2010-2017) and ‘Materials and Methods’. Which one is correct?

 Thank you for your kind comments. It is corrected as suggested by you.

Table 1 would be better placed in the ‘Results session’.

 Thank you for your kind comments. It is corrected as suggested by you.

In the Results session, it would be interesting to have the demographic profile of the interviewees (sex/age/profession/district).

 Thank you for your kind comments. It is corrected as suggested by you.

The second session in ‘Results’ is named ‘Preparations, application and dose-dependence of TFs’, but there is no mention of the dose-dependence in the results.

 Thank you for your kind comments. It (dose dependence) is deleted as suggested by you.

Table 2 should me made horizontally, to increase readability and should be placed in the ‘Quantitative ethnobiology’ session. 

 Thank you for your kind comments. It is corrected as suggested by you.

Also, in this table, what does “voucher number’ stands for? And the numbers that appear after the description of ‘parts used’? This information should be removed or explained in the table’s legend. 

 Thank you for your kind comments. It is corrected as suggested by you. Voucher no. and the numbers that appear after the description of ‘parts used’ are explained in the footnote.

Information on abbreviations is also lacking in table 5.

 Thank you for your kind comments. It is added as suggested by you.

As a suggestion, data presented in table 6 would be more clearly visualized as a Graph.

 Thank you for your kind comments. It is corrected as suggested by you.Table 6 is replaced by

Table 7 would also be better read horizontally.

 Thank you for your kind comments. It is corrected as suggested by you.

The Discussion session, unlike the other session, is rather succinct. I believe the other parts of the manuscript could be more concise and Discussion should be further explored, comparing the found information with other similar articles, performed in other regions.

 Thank you for your kind comments. It is corrected as suggested by you.

Discussion part is further elaborated comparing the found information with other similar articles, performed in other regions.

In line 419, authors state that ‘Most of the snake bites are either dry bites or from non-Poisonous snakes. Still death occurs. There are many instances where patients die only because of fear and psychological shock.’ This seemed odd to me. Are there any references to support this statement?

 Thank you for your kind comments. References are added to support this statement.

In line 494 of the ‘Conclusion’, authors claim that ‘Various conservation strategies have also been described as adopted by the indigenous people in order to use plant resources sustainably.’. I did not find the description of these strategies throughout the text.

 Thank you for your kind comments. This part is deleted and rewritten.

Reviewer #2: 

A broad review regarding the English language, punctuation and spacing should be made. Thank you for your kind comments. We have corrected the language with the best of our ability.

Several references are mentioned incorrectly during the text, such as seen on lines 102, 105 and 109, 

 Thank you for your kind comments. It is corrected as suggested by you.

During introduction, authors stress that PLA2 enzymes are responsible for the inflammatory effects of snake bites, and that plant extracts detain some sort of PLA2 neutralizing capacity. However, activity of PLA2 of both human and porcine origin are also evaluated. These subgroups must be more clearly determined in methodology and results and discussion.

Authors should classify the different Authors also claim to evaluate genotoxicity and cytotoxicity, and that should be clear as an aim of the present study as well. 

 Thank you for your kind comments. It is corrected as suggested by you. We have added portions in the methodology and result sections where the subgroups are more clearly determined. We have also added a number of references in the discussion to classify the different Authors also claim to evaluate genotoxicity and cytotoxicity.

The aim of the study is not clearly outlined by the authors.

 Thank you for your kind comments. The aim of the study is now clearly added at the end of introduction section.

On Table 1, TF4 is explained as masticated by hand, a term with dubious meaning.

 Thank you for your kind comments. It was corrected and replaced.

Table 2 is impossible to read and contains some of the main results from the experiment that should be clearly available to the reader.

 Thank you for your kind comments. The table was reframed to be clearly available to the reader.

Table 7 also needs formatting.

 Thank you for your kind comments. As suggested by the first reviewer we have replaced the data of Table 7 with Figure 7 presenting the same dataset.

Discussion is very superficial, mainly replicating results obtained. It should be rewritten, stressing what each plant extract detains from a biochemical or pharmacological level that corroborates with a lesser or more prominent effect regarding PLA2 neutralization.

 Thank you for your kind comments. The discussion part is elaborated explaining the plant extract from a biochemical or pharmacological level with a number of newly added references which corroborates with a lesser or more prominent effect regarding PLA2 neutralization.

Since no aim of the study was properly established, conclusion is also in need of revision.

 Thank you for your kind comments. We have rewritten the aim of the study and hence the conclusion is also rewritten accordingly.

---

## [Decision Letter · Decision Letter 1]

20 Oct 2020

PONE-D-20-17909R1

An evidence based efficacy and safety assessment of the ethnobiologicals against poisonous and non-poisonous bites used by the tribals of three westernmost districts of West Bengal, India: Anti-phospholipase A2 and genotoxic effects

PLOS ONE

Dear Dr. Malik,

Thank you for submitting your manuscript to PLOS ONE. After careful consideration, we feel that it has merit but does not fully meet PLOS ONE’s publication criteria as it currently stands. Therefore, we invite you to submit a revised version of the manuscript that addresses the points raised during the review process.

We look forward to receiving your revised manuscript.

Kind regards,

Benito Soto-Blanco, DVM, MSc, PhD

Academic Editor

PLOS ONE

Reviewers' comments:

Reviewer's Responses to Questions

**Comments to the Author**

1. If the authors have adequately addressed your comments raised in a previous round of review and you feel that this manuscript is now acceptable for publication, you may indicate that here to bypass the “Comments to the Author” section, enter your conflict of interest statement in the “Confidential to Editor” section, and submit your "Accept" recommendation.

Reviewer #1: (No Response)

2. Is the manuscript technically sound, and do the data support the conclusions?

Reviewer #1: Yes

3. Has the statistical analysis been performed appropriately and rigorously? 

Reviewer #1: N/A

4. Have the authors made all data underlying the findings in their manuscript fully available?

Reviewer #1: Yes

5. Is the manuscript presented in an intelligible fashion and written in standard English?

Reviewer #1: Yes

6. Review Comments to the Author

Reviewer #1: Although the manuscript has been consistently improved from the first submission, there are still some points that should be further addressed prior to publication:

 In Table 1, there are still unexplained numbers, placed after "Disease/ Disorder" and after some words like "ointment

(1070000)". These numbers should be further explained in the legend or footnote. In addition, the units abbreviations described in the TF Composition (eg: Seowra (Streblus asper) leaves: 4-5 in no.) should also be described in the table's footnotes.

 Likewise, for Table 2, numbers placed after "Parts used" (eg: root (12040000)), and "Ailment treated" are not explained neither in the table's legend nor in the footnote.

 The session "Use of ethnobotanicals: toxicity, conservation status and Economic Botany Data Standard" is still a bit confusing, as different subjects are discussed in the same paragraph, without an evident connection between them.

 I am still concerned about the statement in lines 428-429: "There are many instances where patients die only because of fear and psychological shock." I believe that, indeed, the stress caused by a snakebite can cause some harm but not be the responsible for the victim's death. I did not find support to this statement in the respective references. Therefore, I would recommend to rephrase the statement or to remove it from the discussion.

7. PLOS authors have the option to publish the peer review history of their article (what does this mean?). If published, this will include your full peer review and any attached files.

Reviewer #1: No

---

## [Author Response · Author response to Decision Letter 1]

3 Nov 2020

As needed Figures S1, S2 and S3 are included in this version of submission.

Although the manuscript has been consistently improved from the first submission, there are still some points that should be further addressed prior to publication: 

Our answer: Thank you.

In Table 1, there are still unexplained numbers, placed after "Disease/ Disorder" and after some words like "ointment (1070000)". These numbers should be further explained in the legend or footnote. 

In addition, the units abbreviations described in the TF Composition (eg: Seowra (Streblus asper) leaves: 4-5 in no.) should also be described in the table's footnotes. 

Our answer: The numbers used in parenthesis after the diseases/disorders, parts and methods of administration in table 1 and 2 are according to the recommendations given by Cook, 1995 as Economic botany data collection standard (EBDCS), plant parts, body parts and processes, disorders/effects, medicinal applications and non-vertebrate organisms (Master lists of states for Level 3 descriptors) (Economic Botany Data Standard; https://www.kew.org/tdwguses/rptMasterListMain.htm). 3In composition, in no. used after the numbers stands for in number i.e. the number of that plant part used.

This information is added to the footnote in both the tables. 

In addition, the units abbreviations described in the TF Composition (eg: Seowra (Streblus asper) leaves: 4-5 in no.) are also described in the table's footnotes.

Likewise, for Table 2, numbers placed after "Parts used" (eg: root (12040000)), and "Ailment treated" are not explained neither in the table's legend nor in the footnote. 

Our answer: The numbers used in parenthesis after the diseases/disorders, parts and methods of administration in table 1 and 2 are according to the recommendations given by Cook, 1995 as Economic botany data collection standard (EBDCS), plant parts, body parts and processes, disorders/effects, medicinal applications and non-vertebrate organisms (Master lists of states for Level 3 descriptors) (Economic Botany Data Standard; https://www.kew.org/tdwguses/rptMasterListMain.htm). 3In composition, in no. used after the numbers stands for in number i.e. the number of that plant part used.

This information is added to the footnote in both the tables.

The session "Use of ethnobotanicals: toxicity, conservation status and Economic Botany Data Standard" is still a bit confusing, as different subjects are discussed in the same paragraph, without an evident connection between them. 

Our answer: Yes, that is why we removed the irrelevant Economic Botany Data Standard from this section, explaining as footnotes in table 1 and 2 as suggested by the reviewers. This section of the result is now renamed as “Use of ethnobotanicals: toxicity, aspects and conservation status” in accordance to the section in the discussion “Use of plants and animals as drugs: toxicity and conservation aspects”

---

## [Decision Letter · Decision Letter 2]

12 Nov 2020

An evidence based efficacy and safety assessment of the ethnobiologicals against poisonous and non-poisonous bites used by the tribals of three westernmost districts of West Bengal, India: Anti-phospholipase A2 and genotoxic effects

PONE-D-20-17909R2

Dear Dr. Malik,

We’re pleased to inform you that your manuscript has been judged scientifically suitable for publication and will be formally accepted for publication once it meets all outstanding technical requirements.

Kind regards,

Benito Soto-Blanco, DVM, MSc, PhD

Academic Editor

PLOS ONE

Reviewers' comments:

Reviewer's Responses to Questions

**Comments to the Author**

1. If the authors have adequately addressed your comments raised in a previous round of review and you feel that this manuscript is now acceptable for publication, you may indicate that here to bypass the “Comments to the Author” section, enter your conflict of interest statement in the “Confidential to Editor” section, and submit your "Accept" recommendation.

Reviewer #1: All comments have been addressed

2. Is the manuscript technically sound, and do the data support the conclusions?

Reviewer #1: Yes

3. Has the statistical analysis been performed appropriately and rigorously? 

Reviewer #1: Yes

4. Have the authors made all data underlying the findings in their manuscript fully available?

Reviewer #1: Yes

5. Is the manuscript presented in an intelligible fashion and written in standard English?

Reviewer #1: Yes

6. Review Comments to the Author

Reviewer #1: (No Response)

7. PLOS authors have the option to publish the peer review history of their article (what does this mean?). If published, this will include your full peer review and any attached files.

Reviewer #1: No

---

## [Editor Report · Acceptance letter]

16 Nov 2020

PONE-D-20-17909R2 

An evidence based efficacy and safety assessment of the ethnobiologicals against poisonous and non-poisonous bites used by the tribals of three westernmost districts of West Bengal, India: Anti-phospholipase A2 and genotoxic effects 

Dear Dr. Malik:

I'm pleased to inform you that your manuscript has been deemed suitable for publication in PLOS ONE. Congratulations! Your manuscript is now with our production department. 

Kind regards, 

on behalf of

Dr. Benito Soto-Blanco 

Academic Editor

PLOS ONE